# Utility of the Cobas^®^ Plasma Separation Card as a Sample Collection Device for Serological and Virological Diagnosis of Hepatitis C Virus Infection

**DOI:** 10.3390/diagnostics11030473

**Published:** 2021-03-08

**Authors:** Fernando Velásquez-Orozco, Ariadna Rando-Segura, Joan Martínez-Camprecios, Paula Salmeron, Adrián Najarro-Centeno, Àngels Esteban, Josep Quer, María Buti, Tomás Pumarola-Suñe, Francisco Rodríguez-Frías

**Affiliations:** 1Department of Microbiology, Vall d’Hebron Hospital Universitari, Vall d’Hebron Barcelona Hospital Campus, Passeig Vall d’Hebron 119-129, 08035 Barcelona, Spain; fvelasquez@vhebron.net (F.V.-O.); psalmeron@vhebron.net (P.S.); tpumarola@vhebron.net (T.P.-S.); 2Department of Genetics and Microbiology, Universitat Autònoma de Barcelona, 08193 Bellaterra, Spain; frarodri@vhebron.net; 3Liver Pathology Unit, Department of Microbiology and Biochemistry, Vall d’Hebron Hospital Universitari, Vall d’Hebron Barcelona Hospital Campus, Passeig Vall d’Hebron 119-129, 08035 Barcelona, Spain; a.najarro@vhebron.net (A.N.-C.); aesteban@vhebron.net (À.E.); 4Department of Internal Medicine, Vall d’Hebron Hospital Universitari, Vall d’Hebron Barcelona Hospital Campus, Passeig Vall d’Hebron 119-129, 08035 Barcelona, Spain; joan.martinez@vhebron.net (J.M.-C.); mbuti@vhebron.net (M.B.); 5Liver Unit, Vall d’Hebron Institut de Recerca (VHIR), Vall d’Hberon Hospital Universitari, Vall d’Hebron Barcelona Hospital Campus, Passeig Vall d’Hebron 119-129, 08035 Barcelona, Spain; josep.quer@vhir.org; 6Department of Biochemistry, Vall d’Hebron Hospital Universitari, Vall d’Hebron Barcelona Hospital Campus, Passeig Vall d’Hebron 119-129, 08035 Barcelona, Spain; 7CIBER de Enfermedades Hepáticas y Digestivas (CIBEREHD), Instituto de Salud Carlos III, Avenida de Monforte de Lemos 3-5, 28029 Madrid, Spain

**Keywords:** hepatitis C, dried plasma spot, plasma separation card

## Abstract

Diagnosis and clinical management of people infected with hepatitis C virus (HCV) relies on results from a combination of serological and virological tests. The aim of this study was to compare the performance of dried plasma spots (DPS), prepared using the cobas^®^ Plasma Separation Card (PSC), to plasma and serum from venipuncture, for HCV diagnosis. We carried out a prospective study using DPS and paired plasma or serum samples. Serum and DPS samples were analyzed by immunoassay using Elecsys^®^ Anti-HCV II (Roche). Plasma and DPS samples were analyzed using the cobas^®^ HCV viral load and cobas^®^ HCV genotyping tests (Roche). All DPS samples that had high anti-HCV antibody titers in serum were also antibody-positive, as were five of eight samples with moderate titers. Eight samples with low titers in serum were negative with DPS. Among 80 samples with plasma HCV viral loads between 61.5 and 2.2 × 10^8^ IU/mL, 74 were RNA-positive in DPS. The mean viral load difference between plasma and DPS was 2.65 log_10_ IU/mL. The performance of DPS for detection of serological and virological markers of hepatitis C virus infection was comparable to that of the conventional specimen types. However, the limits of detection were higher for DPS.

## 1. Introduction

According to recent estimates, more than 71 million people worldwide are infected with the hepatitis C virus (HCV) [1]. HCV causes both acute and chronic infection. New HCV infections are usually asymptomatic. Therefore, most infected individuals are not aware of their infection. Although around 30% (15–45%) of infected persons spontaneously clear the virus within 6 months of infection without any treatment, the remaining 70% (55–85%) will develop chronic HCV infection carrying a risk of cirrhosis of between 15% and 30% within 20 years [2]. Treatment with direct-acting antivirals can result in cure rates of over 95%, and provides more benefits when the disease is treated in its initial stages [3]. These improvements in therapeutic tools have enabled the World Health Organization (WHO) to propose the objective of HCV eradication by 2030. To reach this objective, it is critical to identify infected individuals, to link them to care and therapy and to avoid onward transmission [4].

Screening for HCV infection is based on the detection of anti-HCV antibodies (Abs). If anti-HCV Abs are detected, the presence of HCV RNA should be determined to identify patients with ongoing infection [5]. This makes the diagnosis and the clinical management of HCV infection complex and requires sophisticated laboratory skills [6]. However, this diagnostic procedure has been significantly simplified with the widespread use of a one-step screening and diagnosis strategy for viremic HCV infection (reflex testing), in which the same specimen is used to determine HCV RNA in anti-HCV-positive patients, eliminating the need for the physician to request a second test and reducing the number of visits [5,7].

Significant scale-up in access to hepatitis testing and treatment will require further simplification of the diagnosis and monitoring process, and methods to facilitate access to testing, especially in decentralized settings and among vulnerable populations worldwide, such as people who inject drugs (PWID) and people in prison [8]. The collection, storage, and transport of frozen serum or plasma, the standard specimen type, is not always possible in these settings. Dried blood spots (DBSs) represent an interesting alternative specimen type that does not require venipuncture and is being increasingly used to facilitate access to serological testing and nucleic acid amplification tests (NAATs) for HIV, hepatitis B and C, and other infectious diseases. DBSs are particularly useful in remote and under-resourced regions with poor access to laboratory services, as well as for large epidemiological surveillance studies [8,9,10]. DBS preparation involves obtaining a whole blood specimen, usually by capillary finger-stick (or heel-prick in infants), and embedding the drops of blood onto filter paper, or by pipetting venous blood onto filter paper. DBS specimens can then be transported by standard means (e.g., regular mail) from remote areas to a laboratory, where testing would take place. The simplicity and relative ease of specimen collection, preparation, transport, and storage make DBS specimens a potential option for serological testing and NAATs in low-resource settings [8].

A device for the collection and stabilization of dried plasma from whole blood specimens has recently been commercialized, the cobas^®^ Plasma Separation Card (PSC) [11]. This collection device shares many features with DBSs, but it has some advantages due to higher volume capacity and easy-to-use transportation and storage, while also providing a state-of-the-art stabilization membrane for storage and transport of samples under conditions of extreme heat and humidity [11]. The PSC is designed for collection and storage of specimens for subsequent in vitro diagnostics examination.

The aim of this study was to evaluate the performance of dried plasma spots (DPSs) prepared using the PSC as a specimen type for serologic and virologic diagnosis of HCV. Some previous studies evaluated the utility of DPS in the diagnosis of hepatitis C, most of which were carried out with Whatman^®^ Protein Saver Cards. A previous study evaluated the utility of the PSC for quantifying HCV RNA [12]. Our study evaluates for the first time the performance of this device for complete microbiological assessment of hepatitis C infection.

## 2. Materials and Methods

### 2.1. Study Design

This study was carried out at Clinical Laboratories of Vall d’Hebron Hospital in Barcelona, Spain, which serves a population of 1.2 million citizens (inpatients and outpatients) in addition to 12 drug abuse and rehabilitation centers and a women’s prison. A prospective case-control study was carried out during October and November 2019. Remnant ethylenediaminetetra-acetic acid (EDTA) anticoagulated venous blood and serum or plasma samples from HCV Ab-positive patients attending the Clinical Laboratories of Vall d′Hebron Hospital were selected regardless of the reason for the request. Because a low number of viremic patients were detected, additional samples were collected from May to July 2020. In this case, remnant EDTA anticoagulated venous blood and serum or plasma samples from patients with detectable HCV RNA were selected.

### 2.2. Sample Preparation

DPSs were prepared by spotting 140 μL of whole EDTA venous blood in each of the 3 spots of the cobas^®^ PSC (Figure 1) and drying at room temperature, followed by storage at room temperature without desiccant for 4 to 21 days before processing.

### 2.3. Serological Testing

Serum samples (350 µL) were processed for serological testing according to the manufacturer’s instructions. For DPS samples, individual dried spots were removed from the cards and incubated overnight (at least 8 h) at 37 °C in 600 μL of universal diluent (Roche, Basel, Switzerland) dedicated to Elecsys serological assays. After the incubation period, we centrifuged the samples at 1300 rpm for 10 min. Before introducing the samples into the analyzer, the entire eluate was transferred to a new tube, leaving the PSC paper in the primary tube. Samples were analyzed with the Elecsys^®^ Anti-HCV II (Roche, Basel, Switzerland) for use on the cobas e601/cobas e602 modules or cobas e801 modules of cobas^®^ 8000 modular analyzer series. If the result of the serological test was positive, another DPS was used to perform the viral load analysis.

### 2.4. Viral Load Testing

Plasma samples (500 µL) were processed for virological testing according to the manufacturer’s instructions. For DPS samples, individual dried spots were removed from the cards and incubated at 56 °C with 950 μL of Specimen Pre-Extraction Reagent (Roche) for 10 min at 1000 rpm on a preheated thermoshaker. Before introducing the samples into the analyzer, the entire eluate was transferred to a new tube, leaving the PSC paper in the primary tube. Samples were analyzed with the cobas^®^ HCV test (Roche, Basel, Switzerland) for use on the cobas 6800 system using the 500 μL processing volume option (overall analytical limit of detection [LoD] was 12.0 IU/mL). No correction factor was used to adjust the viral load from PSC.

### 2.5. HCV Genotyping

Plasma samples (500 µL) were processed for virological testing according to the manufacturer’s instructions. For DPS samples, individual dried spots were removed from the cards and incubated at 56 °C with 950 μL of Specimen Pre-Extraction Reagent (Roche, Basel, Switzerland) for 10 min at 1000 rpm on a preheated thermoshaker. Before introducing the samples into the analyzer, the entire eluate was transferred to a new tube, leaving the PSC paper in the primary tube. Samples were analyzed with the cobas^®^ HCV genotyping (GT) test (Roche, Basel, Switzerland) for use on the cobas^®^ 4800 System. The cobas 4800 system automatically performs the nucleic acid extraction, PCR assay preparation, and transfers samples to 96-well plates, ready for PCR amplification and detection. The cobas^®^ HCV GT is designed for the identification of HCV genotypes 1 to 6 and subtypes 1a and 1b. Excess material was used to perform an RNA extraction with the MagnaPure 24 System using the protocol Pathogen 200 hp 1.0. and high-resolution HCV subtyping using NS5B deep sequencing (MiSeq, Illumina, Sant Diego, CA, USA) and phylogenetic analysis [13].

### 2.6. Statistical Analysis

We performed statistical analysis using Stata 12 (StataCorp LLC, College Station, TX, USA). Performance parameters (sensitivity and specificity) of the anti-HCV Ab test in DPSs were estimated by using the results from serum specimens as a gold standard. Discordant results were submitted to confirmatory serological testing. Viral load results were log-transformed before statistical analysis. Pearson correlation and Bland–Altman analysis were performed to compare viral load in plasma and DPS. Performance parameters (sensitivity and specificity) of the HCV RNA assay in DPS using the European Association for the Study of the Liver (EASL) recommended 1000 IU/mL threshold were estimated using the results from plasma specimens as the gold standard [5].

## 3. Results

### 3.1. Serological Diagnosis

We selected 101 anti-HCV Ab-positive and 50 Ab-negative patients based on Elecsys^®^ Anti-HCV II results in serum. All 50 patients who tested negative in serum also tested negative in DPS. Of the 101 patients who tested positive in serum, 90 also tested positive in DPS; all 85 samples with high (cut-off index >25) HCV Ab titer in serum were positive with DPS (Table 1). Discordant DPS samples were all from patients with low (cut-off index <10; zero of eight, 0%) or medium (cut-off index between 10 and 25; five of eight, 63%) HCV Ab titer (Table 1). These 11 discordant samples belonged to patients who tested negative for HCV RNA in plasma. Three of these samples belonged to patients who were cured, one following treatment with interferon in 2006 and two with spontaneous HCV clearance. The remaining patients with discrepant results had chronic renal failure in substitutive therapy with hemodialysis (three patients) or autoimmune pathologies (three patients), and there was one psychiatric patient and one pregnant woman. To clarify these discrepant results, confirmatory testing using the INNO-LIA^®^ HCV Score (Fujirebio, Tokyo, Japan) was performed in the four cases for which enough sample was available. Confirmation tests were negative in three of these four cases: one patient with autoimmune pathology and two patients in substitutive therapy with hemodialysis. The remaining sample was indeterminate: a patient with autoimmune pathology who had multiple positive anti-HCV determinations, all of them close to the cut-off value. Overall, excluding these four false-positive samples, there was a good agreement in detection of anti-HCV with sensitivity of 92.8% and specificity of 100%.

### 3.2. Determination of HCV RNA Viral Load

Among the 101 anti-HCV Ab-positive samples, only 36 had detectable HCV RNA. Therefore, we identified 44 additional samples with detectable HCV-RNA in plasma, for a total of 80 samples, with viral load in plasma ranging from 61.5 to 2.2 × 10^8^ IU/mL. All 65 patients who tested undetectable in plasma also tested undetectable in DPS. Two patients had plasma viral load lower than 1000 IU/mL, which was not detected with DPS. Among the remaining 78 samples with viral load over 1000 IU/mL, 74 (94.9%) were detectable in DPS. Three of the five samples with plasma between 1000 and 10,000 IU/mL had undetectable RNA in DPS. Another sample with undetectable HCV RNA in DPS had a plasma viral load of 46,800 IU/mL (Table 2). We observed that during the preparation of the DPSs, some of the high-viscosity whole blood clogged the membrane of the PSC. Correlation of viral load results derived from DPS and plasma in samples with detectable RNA levels in both specimens was moderate but linear (slope = 0.808, intercept = −1.45, R^2^ = 0.516; Figure 2A). The mean viral load difference between plasma and DPS was −2.64 log_10_ IU/mL (95% CI: 2.49 to 2.79) (Figure 2B). Overall, excluding samples with viral load below 1000 IU/mL, there was a good agreement in detection of HCV-RNA, with sensitivity of 94.98% and specificity of 100%.

### 3.3. Determination of HCV Genotype

The 44 additional samples with detectable HCV-RNA in plasma were used to evaluate the performance of DPS for HCV genotyping. These samples had plasma viral loads ranging from 61.5 to 2.5 × 10^7^ IU/mL. Three samples could not be amplified for genotyping: the one with viral load below 1000 IU/mL, and two of three samples with viral load between 1000 and 10,000 IU/mL. Genotype assignments were concordant between plasma and DPS for 40 samples (40/41; 97.6%), with only one discordant sample, which was assigned genotype 1b from plasma but mixed genotype 1b and 2 from DPS. We also performed high-resolution HCV subtyping on these samples by deep sequencing, which was successful in 22 of them (53.7%). Viral loads among the samples with successful vs. unsuccessful high-resolution HCV subtyping tests were similar. The subtype assignments of 21/22 samples (95.5%) were concordant, with only one discordant sample assigned genotype 3a from plasma but mixed genotype 1a and 3a from DPS.

## 4. Discussion

For diagnosis of viral infections, viral antigens and nucleic acid, and host Ab against viral antigens, are the main targets. Tests that detect the presence of these analytes and measure their concentrations are most often performed using blood plasma or serum samples collected by venipuncture. However, this kind of sample is not always available. An alternative to venipuncture samples, from which all these tests could be carried out, would be a big help to try to simplify the diagnostic algorithm in areas of the world where the socio-political and economic environment makes it difficult. Multiple rapid immunochromatographic tests for HCV Ab detection are available, which have acceptable but much lower sensitivity and specificity than laboratory-based electrochemiluminescence methodologies. Furthermore, these tests only provide half of the required information and incomplete diagnosis for Ab-positive samples, which require NAAT testing before making the decision to initiate antiviral treatment [14,15]. This second step is often difficult in these complex populations. Point-of-care NAAT tests to support complete HCV diagnosis are also feasible, with very good sensitivity and specificity and a relatively easy interpretation. Point-of-care tests are performed using relatively portable platforms such as those provided by the GeneXpert system [16]. However, high costs make implementation of most of these solutions difficult in settings with limited resources. Furthermore, despite simple management and interpretation, the complete diagnosis process is not supervised by a microbiologist, with a significant risk of suboptimal application. In this regard, DBS, sent to reference laboratories for centralized testing, has been suggested as an alternative diagnostic tool for viral infectious diseases as HCV [10].

DBSs allow HCV diagnosis to be made with capillary blood from digital puncture, allowing them to be prepared without the infrastructure needed for venipuncture. DBS samples can be processed in high-technology analyzers with the most sensitive and specific techniques available in centralized laboratories, with appropriate supervision and interpretation by a microbiologist. A 2017 meta-analysis in which nineteen studies were included showed a pooled sensitivity and specificity of DBS for hepatitis B surface antigen and anti-HCV Ab detection of 98% and 100%, respectively [17]. Moreover, DBS samples can be easily transported to central and reference clinical laboratories by mail, due to the lack of requirement for a cold chain. However, DBS stability for prolonged duration is limited and may not be feasible for use in some situations [18]. Another limitation of DBSs is that they contain whole blood, which includes red blood cells, lymphocytes, and other potentially interfering substances. This fact is relevant because serological and virological methods are usually standardized for use with serum or plasma. The cobas^®^ PSC sample collection device shares many features with DBS, but collects plasma rather than whole blood and offers additional sample stability, further improving on the utility of the DBS. For example, HIV-1 RNA was found to be stable at 45 °C for 3 weeks [11].

Our group is the first, to our knowledge, to evaluate the performance of DPS prepared using PSC as a specimen type for serologic diagnosis of HCV. We have carried out parallel evaluations under controlled laboratory conditions (present study), as well as in real practice [19]. Under controlled laboratory conditions, the sensitivity and specificity of DPS for anti-HCV Ab testing were 92.8% and 100%, respectively. However, all 11 discordant samples had low levels of anti-HCV Ab in conventional serum samples. Based on the clinical data, in nine of the 11 patients, the serum result may be considered false-positive due to the high sensitivity of the reference technique. It should be taken into account that a false-positive result allows for the performance of a confirmatory test, while a false-negative result due to use of a less sensitive technology would not, resulting in loss of the diagnosis. The confirmatory analysis by INNO-LIA^®^ HCV Score in the four available samples from these 11 discordant cases (positive anti-HCV Ab in serum and negative result in DPS) was negative or indeterminate, strongly suggesting that the discordant cases could be truly seronegative. These 11 samples belonged to patients who had no detectable HCV RNA in plasma. Thus, a negative anti-HCV Ab result from PSC would not have led to an erroneous clinical decision, that is, to not treat these patients with antiviral therapy.

Several studies regarding use of DBS for HCV viral analysis showed high correlation between viral loads measured in plasma vs. DBS [12,20]. In our study, we were able to detect HCV RNA in almost all patients (72/73; 98.6%) with viral loads over 10,000 IU/mL. However, HCV RNA was undetectable in DPS from one patient with viral load of 46,800 IU/mL. During the preparation of some DPS samples, we observed that the membrane of the PSC became clogged due to high viscosity of the whole blood sample, restricting the plasma recovery. On the other hand, we only detected the RNA from DPS in 40% (2/5) of the samples with plasma viral loads between 1000 and 10,000 IU/mL, and none (0/2; 0%) of the patients with plasma viral loads below 1000 IU/mL. The mean of the difference between viral loads measured from plasma vs. DPS was 2.65 log_10_ IU/mL (95% CI: −2.50 to −2.79). This difference is at least partially expected, due to the seven-fold smaller volume of plasma (approximately 70 µL, assuming a hematocrit of 50%, compared to 500 µL for plasma). However, the number of samples studied here is still too low to allow generalization of our results, and further studies with more samples with plasma viral load between 1000 and 10,000 are needed.

Data other than viral load results are required for adequate management of chronic HCV infection, including HCV genotyping and subtyping in some cases. In our study, HCV genotyping could be performed from all DPS samples prepared from viremic patients. However, the high-resolution HCV subtyping test was not always successful, and there does not seem to be a relationship between HCV viral load and the success of this technique. Discordant genotype results were observed in two cases. These discordances could be explained, especially in the case of discordance in high-resolution HCV subtyping, by cross-contamination during the manual processing of the sample, introducing low amounts of RNA during PSC handling [13,21].

Our study had several limitations. First, the study design did not allow for the possibility of re-testing samples to identify HCV Ab-false-positive results for all patients. The DPSs were prepared the same day as the analysis of the paired plasma or serum sample, but because of the drying process, they were not analyzed until between 4 and 21 days later. In our high-activity laboratory (more than 6000 patient samples per day), samples are usually discarded within three days of their processing. Therefore, some of the original samples used to prepare PSC were not available for confirmatory testing. Second, patients with low plasma viral loads were not well represented due to the prospective study design, making it difficult to establish the lower limit of quantitation or limit of detection (LOD) of viral load measurement using DPS. Third, the stability of PSC was not assessed in the study even though samples were processed between 4 and 21 days.

## 5. Conclusions

In conclusion, DPSs are useful as an alternative to plasma or serum from venipuncture blood. This is important in remote regions, but also in some institutions where blood collection services are not always available (e.g., for PWID). DPSs have the advantage of minimizing the possible interference from elements in whole blood on the DBS. Furthermore, HCV serological studies can be performed using DPS. However, the LOD of the techniques employed (serology and NAAT) are affected by the lower amount of starting material and dilution of the sample necessary for its reconstitution compared to plasma or serum. Nonetheless, HCV RNA assays with an LOD ≤1000 IU/mL can be used to provide broad, affordable access to HCV diagnosis and care in low- and middle-income countries. The value of this assay is limited for diagnosis of HCV infection in current or former injection drug users, and probably for prison inmates, due to 1.9% to 6.6% of these patients having a viral load under 1000 IU/ mL [22]. Although HCV elimination has a cost and a few patients will inevitably be missed to allow the others to be found and treated, this fact could be minimized by the modification of the DPS reconstitution process and the use of alternative protocols with higher processing volume [12]. On the other hand, the endpoint of therapy is an sustained virologic response (SVR), defined by undetectable HCV RNA in serum or plasma 12 weeks (SVR12) or 24 weeks (SVR24) after the end of therapy, as assessed by a sensitive molecular method with a lower limit of detection, ≤15 IU/mL. In settings where sensitive HCV RNA assays are not available and/or not affordable, a qualitative assay with a low limit of detection (≤1000 IU/mL) can be used to assess virological response, in which case the response should be assessed at week 24 post-treatment [5].

## Figures and Tables

**Figure 1 diagnostics-11-00473-f001:**
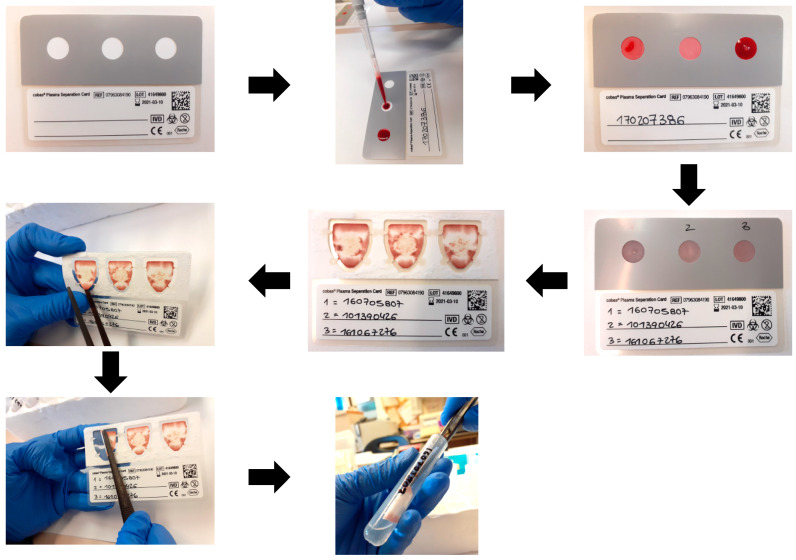
Processing of dried plasma spots using the cobas^®^ Plasma Separation Card.

**Figure 2 diagnostics-11-00473-f002:**
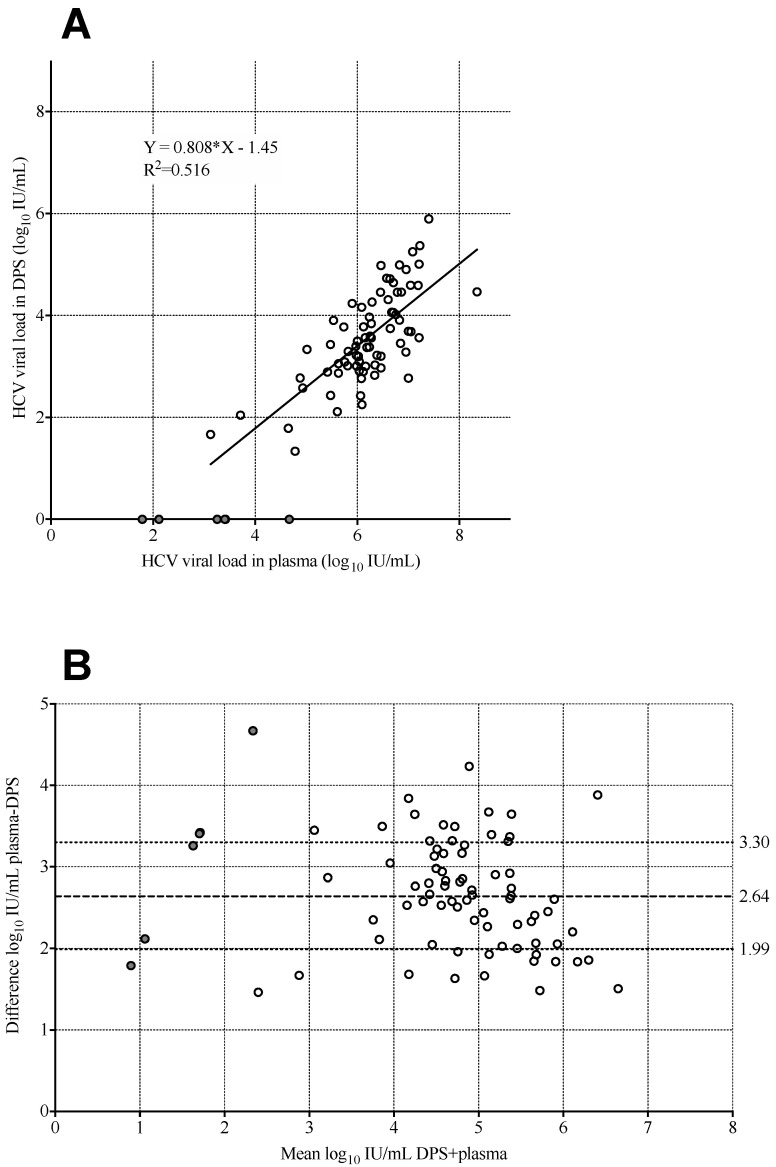
Correlation (**A**) and Bland−Altman plot (**B**) between dried plasma spot (DPS) and plasma samples (*n* = 80). Data points with undetectable HCV RNA (*n* = 6) are shown, shaded in grey; these data were not used for the calculation of the correlation parameters or for the calculation of mean difference. The mean difference between plasma and DPS is shown in a dashed line, and the dotted lines indicate mean plus or minus the standard deviation.

**Table 1 diagnostics-11-00473-t001:** Hepatitis C virus (HCV) serological test results.

Anti-HCV Ab Cut-Off Index in Serum (N)		HCV Ab Detection in PSC	
		Positive ^1^	Negative ^1^
Negative	<0.9 (50)	0	50
Positive	<10 (8)	0	8 ^2^
	10–25 (8)	5	3
	>25 (85)	85	0

^1^ positive: cut-off index >1.0; negative: cut-off index <0.9. ^2^ four of these samples were considered false-positive and excluded from the sensitivity calculation. Ab: antibody; PSC: Plasma Separation Card.

**Table 2 diagnostics-11-00473-t002:** HCV viral load results.

HCV RNA Viral Load in Plasma (N)	HCV RNA Detection in PSC	
	HCV RNA Detected ^1^	HCV RNA Not Detected
HCV RNA not detected	0	65
Lower than 1000 IU/mL	0	2
Over than 1000 IU/mL	74	4

^1^ HCV RNA detected: greater than or equal to 15 IU/mL. No correction factor was used to adjust the viral load in PSC.

## Data Availability

The data presented in this study are available on request from the corresponding author. The data are not publicly available.

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
