# Peer review of "Utility of the Cobas® Plasma Separation Card as a Sample Collection Device for Serological and Virological Diagnosis of Hepatitis C Virus Infection"

_diagnostics, 2021, doi:10.3390/diagnostics11030473_

Round 1
Reviewer 1 Report
This is a relevant study assessing the diagnostic accuracy of the Cobas Plasma Separation Card (PSC) for serological and virological testing. Although the study sample size is very small, it is relevant as only few studies have evaluated the cobas PSC for HCV testing, therefore, the study merits dissemination. The manuscript however requires improvement in its current form prior to publication.
Comments:
1. Authors state that this is a prospective study; however, in the Methods section there isn't any description how participants were selected and neither provide the dates when the study was conducted in terms of participant selection and testing procedures. It is explained that remnant specimens were used and positive and negative ones were purposely selected for the experiments, which gives the impression that this is rather a case-control study. Also in the results section on serological diagnosis it is mention that one sample was from a patient who had received treatment back in 2006. Therefore authors should provide more clarity about the study type.
2. Ethical consideration are lacking in the paper e.g. there is no any mention if the study protocol was approved by an ethics committee and whether participants provided written consent for the use of their remnant specimens for research purposes. Please clarify.
3. Description of the participants recruited is lacking e.g. a table with basic sociodemographic characteristics is necessary. Were patients attending the Vall D'Hebron Hospital seeking a diagnosis for HCV or for follow-up of HCV treatment? Or for other reasons?
4. Statistical analysis: for PSC viral load testing, please mention that VL results were log-transformed before statistical analysis. Also, the Bland-Altman analysis is not mentioned in the statistical methods but described in the results. I strongly suggest that authors also conduct sensitivity/specificity analysis of PSC viral load testing vs plasma using the EASL-recommended 1,000 IU/mL threshold as this is more informative for clinicians. Please also mention what was the threshold used with to deem a sample detectable HCV RNA or undetectable HCV RNA using the cobas 6800 system with a 500 uL sample input. Please also mention whether any correction factor was used for PSC results to account for the lower volume.
5. Results on serological testing: Why the 50 Ab-negative results are excluded from Table 1? The contingency table should include negative and positive results with the serum gold standard, not only positive results.
6. Results on virological testing: correlation plot in Fig 2: please include the total number of paired samples in the legend, (n=80)? Bland-Altman plot in Fig 3: please include the total number of paired samples in the legend, (n=80)? Please also include in the graph the values for the mean difference (dashed line) and 95% limits of agreement (dotted lines). Also report LOA in the main text (Line 167). As mention earlier, I strongly encourage authors to conduct sensitivity/specificity analysis using the 1,000 IU/threshold, like it was done for serological testing. For the sake of space, I would use a combined Figure containing the correlation and Bland-Altman plots to give space for a contingency table for sensitivity/specificity of DPS vs plasma viral load testing.
7. Determination of HCV genotype: why only 44 detectable HCV RNA samples were used? Why not the total 80 detectable samples?
8. Discussion section
– Lines 192 to 210: I feel this is somehow repetition of the introduction/background, therefore I would remove it.
– Lines 225–236: I think in the paragraph no reference is made to previous studies using the PSC for serological testing. If this is the first study to assess this, it should be highlighted.
– Lines 242–244: although the sample size it too small to make conclusions, it is surprising that PSC did not perform well in low-level viremia samples e.g. between 1,000 and 10,000 IU/mL. It is worth highlighting here that this contradicts Roche claims of a limit of detection of 866 IU/mL (95% CI: 698-1153) using the standard 950uL SPER protocol. Also worth mentioning that Roche has developed an optimised 1,300uL ASAP protocol with an improved LOD reaching 408 IU/mL (95% CI: 338–544) and that additional real-world studies merit evaluation.
– Lines 258-266: there are additional limitations of the study that need to be mentioned: e.g. fingerprick samples were not evaluated, which could increase the utility of PSC among PWID where access to venous blood may be challenging. Also, the stability of PSC was not assessed in the study even though samples were processed between 4 and 21 days. I think it is worth also discussing whether PSC can be used for monitoring purposes e.g. to determine SVR12. Arguably, its use for SVR12 may be limited given the LOD of PSC and the low-level viremia in participants experiencing treatment failure, so its use may be reserved only for diagnostic purposes. This is an important implication for its adoption.
Author Response
This is a relevant study assessing the diagnostic accuracy of the Cobas Plasma Separation Card (PSC) for serological and virological testing. Although the study sample size is very small, it is relevant as only few studies have evaluated the cobas PSC for HCV testing, therefore, the study merits dissemination. The manuscript however requires improvement in its current form prior to publication.
Comments:
- Authors state that this is a prospective study; however, in the Methods section there isn't any description how participants were selected and neither provide the dates when the study was conducted in terms of participant selection and testing procedures. It is explained that remnant specimens were used and positive and negative ones were purposely selected for the experiments, which gives the impression that this is rather a case-control study. Also in the results section on serological diagnosis it is mention that one sample was from a patient who had received treatment back in 2006. Therefore authors should provide more clarity about the study type. [Thanks for your comment. We agree with the reviewer. The clarification has been included in Material and Methods section following the reviewer suggestion] [Lines 98-107].
- Ethical consideration are lacking in the paper e.g. there is no any mention if the study protocol was approved by an ethics committee and whether participants provided written consent for the use of their remnant specimens for research purposes. Please clarify. [Thanks for your comment. This study was performed with leftover samples and didn’t require additional blood samples or personal data. The collected samples were not stored after evaluation. For all this reason we request to the ethic committe the dispense and will send it to you as soon as possible.]
- Description of the participants recruited is lacking e.g. a table with basic sociodemographic characteristics is necessary. Were patients attending the Vall D'Hebron Hospital seeking a diagnosis for HCV or for follow-up of HCV treatment? Or for other reasons? [Thanks for your comment. The clinical laboratories at University Hospital Vall d’Hebron in Barcelona, Spain serve a population of 1.2 million citizens (inpatients and outpatients) in addition to 12 drug abuse and rehabilitation centers and a women’s prison. The objective of this study was evaluate the accuracy of the PSC in serological and viralogical diagnosis of hepatitis C. For this reason remnant EDTA anti-coagulated venous blood and serum samples from HCV Ab-positive patients attending the Clinical Laboratories of Vall d'Hebron Hospital were selected regardless of the reason for the request.]
- Statistical analysis: for PSC viral load testing, please mention that VL results were log-transformed before statistical analysis. Also, the Bland-Altman analysis is not mentioned in the statistical methods but described in the results. [Thank you for your comment. We agree with the reviewer. The clarification has been included in Material and Methods section following the reviewer suggestion] [Lines 152-153] I strongly suggest that authors also conduct sensitivity/specificity analysis of PSC viral load testing vs. plasma using the EASL-recommended 1,000 IU/mL threshold as this is more informative for clinicians. [We agree with the reviewer. Calculation and performance parameters of HCV RNA assay in DPS have been included in Materials and Methods and Results, respectively, following the reviewer suggestion] [Lines 154-156]. Please also mention what was the threshold used with to deem a sample detectable HCV RNA or undetectable HCV RNA using the cobas 6800 system with a 500 uL sample input. Please also mention whether any correction factor was used for PSC results to account for the lower volume.[Thank you for your comment. We agree with the reviewer. The clarification has been included in Material and Methods section following the reviewer suggestion] [Lines 134-135].
- Results on serological testing: Why the 50 Ab-negative results are excluded from Table 1? The contingency table should include negative and positive results with the serum gold standard, not only positive results. [Thank you for your comment. We agree with the reviewer. The 50 Ab-negative results has been included in the Table 1]
- Results on virological testing: correlation plot in Fig 2: please include the total number of paired samples in the legend, (n=80)? Bland-Altman plot in Fig 3: please include the total number of paired samples in the legend, (n=80)? Please also include in the graph the values for the mean difference (dashed line) and 95% limits of agreement (dotted lines). Also report LOA in the main text (Line 167). [We agree with the reviewer. Figure have been modified] As mention earlier, I strongly encourage authors to conduct sensitivity/specificity analysis using the 1,000 IU/threshold, like it was done for serological testing. [We agree with the reviewer. Calculation and performance parameters of HCV RNA assay in DPS have been included in Materials and Methods and Results, respectively, following the reviewer suggestion] [Lines 193-195]. For the sake of space, I would use a combined Figure containing the correlation and Bland-Altman plots to give space for a contingency table for sensitivity/specificity of DPS vs. plasma viral load testing. [We agree with the reviewer. Figure has been modified and table 2 has been included]
- Determination of HCV genotype: why only 44 detectable HCV RNA samples were used? Why not the total 80 detectable samples?[Thank for your comment. Remnant EDTA anti-coagulated venous blood and serum/plasma samples from HCV Ab-positive patients attending the Clinical Laboratories of Vall d'Hebron Hospital were selected. Many of these samples didn’t have RNA-VHC requested. EDTA anti-coagulated venous bloods were collected to perform RNA-VHC as reflex testing, and DPS. But the samples volume is limitant and there are not sufficient volumes to perform HCV genotype in plasma. In a second stage remnant EDTA anti-coagulated venous blood and plasma samples from detectable HCV RNA patients were selected. In these cases there are sufficient volumes to perform HCV genotype to original sample.]
- Discussion section
– Lines 192 to 210: I feel this is somehow repetition of the introduction/background, therefore I would remove it. [Thanks for your comment. But we consider important to maintain this information to argue the discussion]
– Lines 225–236: I think in the paragraph no reference is made to previous studies using the PSC for serological testing. If this is the first study to assess this, it should be highlighted. [Thank you for your comment. We agree with the reviewer. The clarification has been included in Discussion section following the reviewer suggestion] [Lines 250-251].
– Lines 242–244: although the sample size it too small to make conclusions, it is surprising that PSC did not perform well in low-level viremia samples e.g. between 1,000 and 10,000 IU/mL. It is worth highlighting here that this contradicts Roche claims of a limit of detection of 866 IU/mL (95% CI: 698-1153) using the standard 950uL SPER protocol. Also worth mentioning that Roche has developed an optimised 1,300uL ASAP protocol with an improved LOD reaching 408 IU/mL (95% CI: 338–544) and that additional real-world studies merit evaluation. [Thank for your comment. It is important highlight that Roche claims of a limit of detection of 866 IU/ml using the standard 950uL SPER protocol, but the difference with this study is that they worked with plasma samples and not with whole blood samples.]
– Lines 258-266: there are additional limitations of the study that need to be mentioned: e.g. fingerprick samples were not evaluated, which could increase the utility of PSC among PWID where access to venous blood may be challenging. Also, the stability of PSC was not assessed in the study even though samples were processed between 4 and 21 days. I think it is worth also discussing whether PSC can be used for monitoring purposes e.g. to determine SVR12. Arguably, its use for SVR12 may be limited given the LOD of PSC and the low-level viremia in participants experiencing treatment failure, so its use may be reserved only for diagnostic purposes. This is an important implication for its adoption. [Thank you for your comment. We agree with the reviewer. The clarification has been included in Discussion and Conclusion section following the reviewer suggestion] [Lines 293-297 and Lines 312-317].
Reviewer 2 Report
This manuscript provides important insight into the utility of the Cobas Plasma Separation Card as a sample collection device for serological and virological diagnosis of Hepatitis C Virus infection. It is well written and comprehensive. There are a few minor revisions and one major that would be beneficial to apply, as listed below;
Introduction
1: Clarify dried plasma spot usage. Why do you want to investigate their utility? For instance, you have outlined DBS advantages for remote settings and marganalised populations. Expand further on the idea of dried plasma spots as an alternative to DBS (line 81-82) and in particular the regulatory benefits with using a registered sample type over research use only (often the case for DBS which does not allow for clinical use, in most higher income countries).
2. Are there any other studies that have explored DPS utility or is this novel research? This should be clarified (with references if appropriate)
Materials and Methods
3: It would be useful to include sensitivity and specificity data for HCV VL analysis from the PSC (dried plasma spot). Is it possible to test a negative cohort for specificity data?
Results
4: Following on from point 3 include sensitivity and specificity results for HCV VL, including a 2x2 table.
5: On line 166 you should state that the Pearson's correlation showed only a modest/ moderate correlation (R2= 0.516).
6: Ensure the same format is used for representing viral load; i.e. use Log IU/mL or whole number VL but do not mix the two. Example in abstract below.
"Among 80 samples with plasma HCV viral loads between 61.5 to 2.2x108IU/ml, 74 were RNA 38positive in DPS. The mean viral load difference between plasma and DPSwas 2.65 log10IU/ml"
Discussion
7: Within the limitations it should be mentioned that the DPS samples were manufactured within a laboratory setting. Future work should consider analysing clinically collected (real-world) samples.
Author Response
This manuscript provides important insight into the utility of the Cobas Plasma Separation Card as a sample collection device for serological and virological diagnosis of Hepatitis C Virus infection. It is well written and comprehensive. There are a few minor revisions and one major that would be beneficial to apply, as listed below.
Introduction
1: Clarify dried plasma spot usage. Why do you want to investigate their utility? For instance, you have outlined DBS advantages for remote settings and marginalized populations. Expand further on the idea of dried plasma spots as an alternative to DBS (line 81-82) and in particular the regulatory benefits with using a registered sample type over research use only (often the case for DBS which does not allow for clinical use, in most higher income countries). [Thanks for your comment. We agree with the reviewer and the Introduction section has been rewritten in attempt to highlight the utility of DPS] [Lines 81-88].
- Are there any other studies that have explored DPS utility or is this novel research? This should be clarified (with references if appropriate). [Thanks for your comment. We agree with the reviewer and the Introduction section has been rewritten in attempt to highlight the utility of DPS] [Lines 91-96].
Materials and Methods
3: It would be useful to include sensitivity and specificity data for HCV VL analysis from the PSC (dried plasma spot). [We agree with the reviewer. Calculation and performance parameters of HCV RNA assay in DPS have been included in Materials and Methods and Results, respectively, following the reviewer suggestion] [Lines 154-156]. Is it possible to test a negative cohort for specificity data? [We agree with the reviewer. We also analyzed the viral load of all anti-HCV Ab positive samples that had undetectable HCV RNA in plasma. This information has been included in Results section following the reviewer suggestion] [Lines 184-185]
Results
4: Following on from point 3 include sensitivity and specificity results for HCV VL, including a 2x2 table. [We agree with the reviewer. A table has been included in Results section following the reviewer suggestion]
5: On line 166 you should state that the Pearson's correlation showed only a modest/ moderate correlation (R2= 0.516). [We agree with the reviewer. The clarification has been included in Results section following the reviewer suggestion] [Lines 192]
6: Ensure the same format is used for representing viral load; i.e. use Log IU/mL or whole number VL but do not mix the two. Example in abstract below: "Among 80 samples with plasma HCV viral loads between 61.5 to 2.2x108IU/ml, 74 were RNA positive in DPS. The mean viral load difference between plasma and DPS was 2.65 log10 IU/ml". [Thanks for your comment. But we consider important to maintain both formats. Whole number VL helps to understand the results and the comparison with EASL-recommended 1,000 IU/mL threshold. Nevertheless, changes in viral load are usually reported as log change. The alternative to express this difference is fold change but are more difficult to interpret.]
Discussion
7: Within the limitations it should be mentioned that the DPS samples were manufactured within a laboratory setting. Future work should consider analyzing clinically collected (real-world) samples. [We agree with the reviewer. The clarification has been included in Discussion section following the reviewer suggestion] [Lines 293-297]
Round 2
Reviewer 1 Report
All the points raised were adequately addressed by the authors, it is a very useful and informative study which merits dissemination.
Author Response
Ethic committee approval is included in the manuscript